# Development of a SFTSV DNA vaccine that confers complete protection against lethal infection in ferrets

Jeong-Eun Kwak [1,7], Young-Il Kim[2,7], Su-Jin Park[2], Min-Ah Yu[2], Hyeok-Il Kwon[2], Sukyeong Eo[1], Tae-Shin Kim[3], Joon Seok[3], Won-Suk Choi [2], Ju Hwan Jeong [2], Hyojin Lee[4], Youngran Cho[4], Jin Ah Kwon[4], Moonsup Jeong[4], Joel N. Maslow[4], Yong-Eun Kim[5], Haili Jeon[5], Kee K. Kim[5], Eui-Cheol Shin[1,3], Min-Suk Song[2], Jae U. Jung [6], Young Ki Choi [2] & Su-Hyung Park [1,3]

Although the incidence of severe fever with thrombocytopenia syndrome virus (SFTSV) infection has increased from its discovery with a mortality rate of 10–20%, no effective vaccines are currently available. Here we describe the development of a SFTSV DNA vaccine, its immunogenicity, and its protective efficacy. Vaccine candidates induce both a neutralizing antibody response and multifunctional SFTSV-specific T cell response in mice and ferrets. When the vaccine efficacy is investigated in aged-ferrets that recapitulate fatal clinical symptoms, vaccinated ferrets are completely protected from lethal SFTSV challenge without developing any clinical signs. A serum transfer study reveals that anti-envelope antibodies play an important role in protective immunity. Our results suggest that Gn/Gc may be the most effective antigens for inducing protective immunity and non-envelope-specific T cell responses also can contribute to protection against SFTSV infection. This study provides important insights into the development of an effective vaccine, as well as corresponding immune parameters, to control SFTSV infection.

[1] Biomedical Science and Engineering Interdisciplinary Program, Korea Advanced Institute of Science and Technology, Daejeon 34141, Republic of Korea. [2] Department of Microbiology, College of Medicine and Medical Research Institute, Chungbuk National University, Cheongju 28644, Republic of Korea. [3] Graduate School of Medical Science and Engineering, Korea Advanced Institute of Science and Technology, Daejeon 34141, Republic of Korea. [4] GeneOne Life Science, Inc., Seoul 06060, Republic of Korea. [5] Department of Biochemistry, Chungnam National University, Daejeon 34134, Republic of Korea. [6] Department of Molecular Microbiology and Immunology, Keck School of Medicine, University of Southern California, Los Angeles, CA 90033, USA. [7] These authors contributed equally: Jeong-Eun Kwak, Young-Il Kim. Correspondence and requests for materials should be addressed to Y.K.C. (email: choiki55@chungbuk.ac.kr) or to S.-H.P. (email: park3@kaist.ac.kr)

Severe fever with thrombocytopenia syndrome (SFTS) first reported in 2012 is a newly emerging tick-borne infectious disease, endemic to China, South Korea, and Japan[1,2], caused by the SFTS virus (SFTSV) belonging to the genus Banyangvirus in the family Phenuiviridae of the order Bunyavirales[3], which is a single-stranded negative-sense RNA virus with three genomic segments, namely L, M, and S[4-7]. Similar to other bunyaviruses, the L segment encodes the viral RNA-dependent RNA polymerase, the M segment encodes the two viral envelope glycoproteins (GPs) Gn and Gc, and the S segment encodes a nucleocapsid protein (N) and non-structural proteins (NSs)[8]. SFTSV is an arbovirus transmitted by the *Haemaphysalis longicornis* tick as the predominant vector[4], as well as by the *Rhipicephalus microplus* tick and others[9]. It can also be transmitted through direct contact with blood and other body fluids from infected individuals[10,11]. The clinical manifestation of SFTS is characterized by fever, thrombocytopenia, and leukocytopenia, as well as vomiting, diarrhea, and multi-system organ failure often accompanied by hemorrhage. Early mortality rates for SFTS were 30%[8] although more recent data from regional health agencies shows rates of 10–20%. The incidence of SFTS has rapidly increased from 2012 to 2018[8,12]. The spread of the tick vector to North America increases the potential for outbreaks of the disease beyond the Far East Asia. Therefore, the World Health Organization (WHO) has included SFTSV in its list of priority target pathogens requiring urgent attention[13].

There is currently no vaccine available to prevent SFTS. Thus, it is of high priority to develop and evaluate potential vaccines to control and halt the spread of this rapidly emerging infectious agent. Since correlates of protective immunity are unknown, the relative roles of T- and B-cell responses are not well defined, which hinder the development of an effective vaccine for SFTSV. The use of animal models that can effectively mirror human infection is necessary to adequately evaluate vaccine efficacy in vivo vaccine[8,14]. Although several lethal SFTSV infection models have been established using immunocompromised mice[15-17], these immunocompromised mouse models do not exhibit a normal antiviral immune response. We recently established an immunocompetent animal model using aged-ferrets (> 4-years-old, *Mustela putorius furo*) as a model of lethal SFTSV infection, which accurately mimics the clinical manifestations of SFTSV infection—including high fever, severe thrombocytopenia, high viral load in the blood, and severe body weight loss—and exhibits a > 90% mortality rate[18].

Advantages of DNA vaccines over traditional vaccines include the relative ease of development and the ability to induce broad immunity to multiple antigens, in addition to stimulating both T cell and antibody immunity, which make them suitable for the development of vaccines against emerging pathogens[19-21]. Here, we describe the development of a DNA vaccine against SFTSV, and demonstrate its immunogenicity and protection against lethal SFTSV infection in our recently established ferret challenge model.

## Results and discussion

**Immunogenicity of SFTSV DNA vaccine candidates in mice.** DNA vaccines were constructed that encode full-length Gn, Gc, N, NS, and the RNA-dependent RNA polymerase (RdRp) genes of SFTSV based on sequences of 31 clinical isolates isolated from patients in China, Korea, and Japan that were available in GenBank (Supplementary Table 1). The consensus sequences were cloned into the modified pVax1 expression vector as described in the methods (Fig. 1a). We examined the expression of Gn, Gc, N, NSs, or RdRp from each DNA plasmid using western blot assays (Fig. 1b).

To investigate the immunogenicity of SFTSV candidate vaccines, we first examined SFTSV-specific T cell and antibody responses in mice. BALB/c mice were immunized with SFTSV DNA plasmids (pVax1-Gn, pVax1-Gc, pVax1-N, pVax1-NSs, and pVax1-RdRp) via intramuscular injection followed by in vivo electroporation[22] at the site of delivery twice at 3-week intervals (Fig. 1c and Supplementary Fig. 1a). Control mice were immunized with a control plasmid backbone (pVax1). When splenocytes from the immunized mice were stimulated with overlapping peptide (OLP) pools corresponding to SFTSV proteins to examine vaccine-induced SFTSV-specific T-cell responses, interferon (IFN)-γ ELISPOT assays showed that SFTSV DNA vaccines could elicit robust T-cell responses against all five SFTSV antigens that was significantly increased with in vivo electroporation further (Fig. 1d). When BALB/c mice ($n = 3$/each DNA plasmid) were immunized with individual DNA plasmid encoding Gn, Gc, N, NSs, or RdRp followed by in vivo electroporation, IFN-γ ELISPOT assays showed that strong vaccine-induced T-cell responses to desired antigen were induced (Supplementary Fig. 2a).

To further characterize vaccine-induced CD8$^+$ T cells, we generated a dextramer to detect vaccine-induced CD8$^+$ T cells to NSs antigen because NSs-specific T-cell response was most vigorous in immunized mice as shown in Fig. 1d. The minimal epitope sequence (NSs$_{247-255}$, YPYLMAHYL) for high-affinity binding to mouse MHC class I, H-2L$^d$ was identified using in silico NetMHC 4.0 analysis and IFN-γ ELISPOT assays (Supplementary Fig. 2b–d). This was then used for the synthesis of an H-2L$^d$ dextramer (Fig. 1e). Phenotypic analysis of vaccine-induced CD8$^+$ T cells detected by the H-2L$^d$ NSs$_{247-255}$ dextramer revealed that SFTSV DNA vaccination with in vivo electroporation could generate a long-lasting effector memory (CD44$^+$CD62L$^-$) CD8$^+$ T-cell response (Fig. 1e and Supplementary Fig. 2e). Furthermore, we found that SFTSV-specific CD8$^+$ T cells induced by DNA vaccination with in vivo electroporation were multifunctional, as a high frequency of CD8$^+$ T cells produced multiple cytokines in mice immunized with DNA vaccines with in vivo electroporation (Supplementary Fig. 3). Next, we investigated anti-SFTSV humoral immune responses in vaccinated mice by performing an in vitro neutralizing assay (the 50% focus reduction neutralization test, FRNT$_{50}$) using sera isolated 3 weeks after the boost immunization. As shown in Fig. 1f, vaccination with in vivo electroporation yielded a significant neutralization response to SFTSV CB1/2014 strain (genotype B). Collectively, our results suggest that SFTSV DNA vaccines with in vivo electroporation elicits both a robust multifunctional SFTSV-specific T-cell response and a strong neutralizing antibody response in mice.

**Vaccine is protective against lethal infection in ferrets.** We have recently developed an age-dependent ferret model of SFTSV infection and pathogenesis that fully recapitulates the clinical manifestations of human infections[18]. While young adult ferrets (<2-years-old) show no clinical symptoms and mortality, SFTSV-infected aged-ferrets (>4-years-old) demonstrate severe thrombocytopenia, reduced white blood cells, and high fever with 93% mortality rate[18]. To evaluate whether the SFTSV DNA vaccines can induce protective immunity against lethal virus challenge, we assessed a combination of vaccine candidates in the lethal SFTSV infection model using aged-ferrets that we recently established[18]. According to the schedule presented in Fig. 2a and Supplementary Fig. 1b, eight naive aged-ferrets (>4-years-old, V1–V8) were immunized with a mixture of all five SFTSV DNA vaccines (pVax1-Gn, pVax1-Gc, pVax1-N, pVax1-NSs, and pVax1-RdRp) three times at 2-week intervals via intradermal

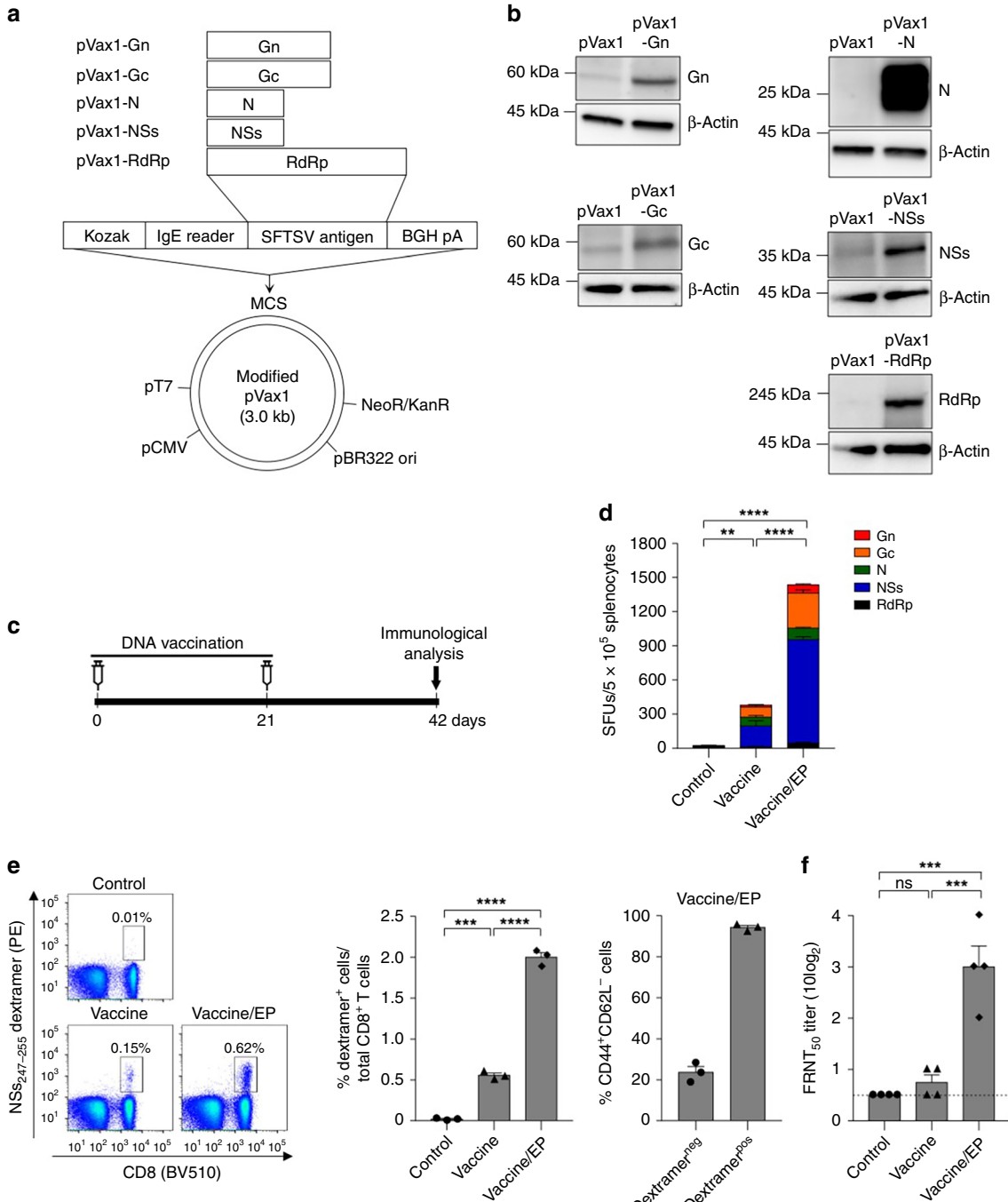

**Fig. 1** Immunogenicity of SFTSV DNA vaccines in BALB/c mice. **a** Diagram of SFTSV DNA vaccines comprising five plasmids encoding SFTSV antigens derived from the common sequence of 31 SFTSV strains. **b** Western blot analysis of SFTSV DNA vaccine expression in 293T cells. **c** Immunization schedule (Supplementary Fig. 1a shows a detailed timeline of vaccination and immune analysis). **d** T-cell immunogenicity of SFTSV DNA vaccines in BALB/c mice. IFN-γ ELISpot assays were performed to detect antigen-specific IFN-γ-producing T cells by stimulating splenocytes from immunized mice with overlapping peptide (OLP) pools. Data (mean ± s.e.m.) represent the average number of spot-forming units (SFUs) per $5 \times 10^5$ splenocytes. **e** Frequency of effector memory (CD44+CD62L−) CD8+ T cells among SFTSV $NSs_{247-255}$-specific CD8+ T cells. Splenocytes from vaccinated mice were stained with H-2L$^d$ $NSs_{247-255}$ dextramer to identify SFTSV $NSs_{247-255}$-specific CD8+ T cells. Then the frequency of CD44+CD62L−CD8+ T cells in this population was examined by multi-color FACS analysis. **f** Neutralizing antibody response to SFTSV CB1/2014 strain (genotype B) generated by SFTSV DNA vaccines in BALB/c mice. In anti-sera collected from vaccinated mice, the amount of neutralizing antibody against SFTSV was determined based on $FRNT_{50}$. Error bars indicate the mean ± s.e.m. Statistical significance was determined by one-way ANOVA test with Tukey correction. **$p < 0.01$; ***$p < 0.001$; ****$p < 0.0001$. Source data are provided as a Source Data file

injection, which was followed by in vivo electroporation at the site of delivery. Another seven naive ferrets of the same ages, comprising the control group (C1–C7), were immunized with the control plasmid backbone (modified pVax1) followed by in vivo electroporation. Two weeks after the last vaccination, the ability of SFTSV DNA vaccines to induce T cell and antibody responses in these vaccinated ferrets was investigated by performing an IFN-γ ELISPOT assay and in vitro neutralization

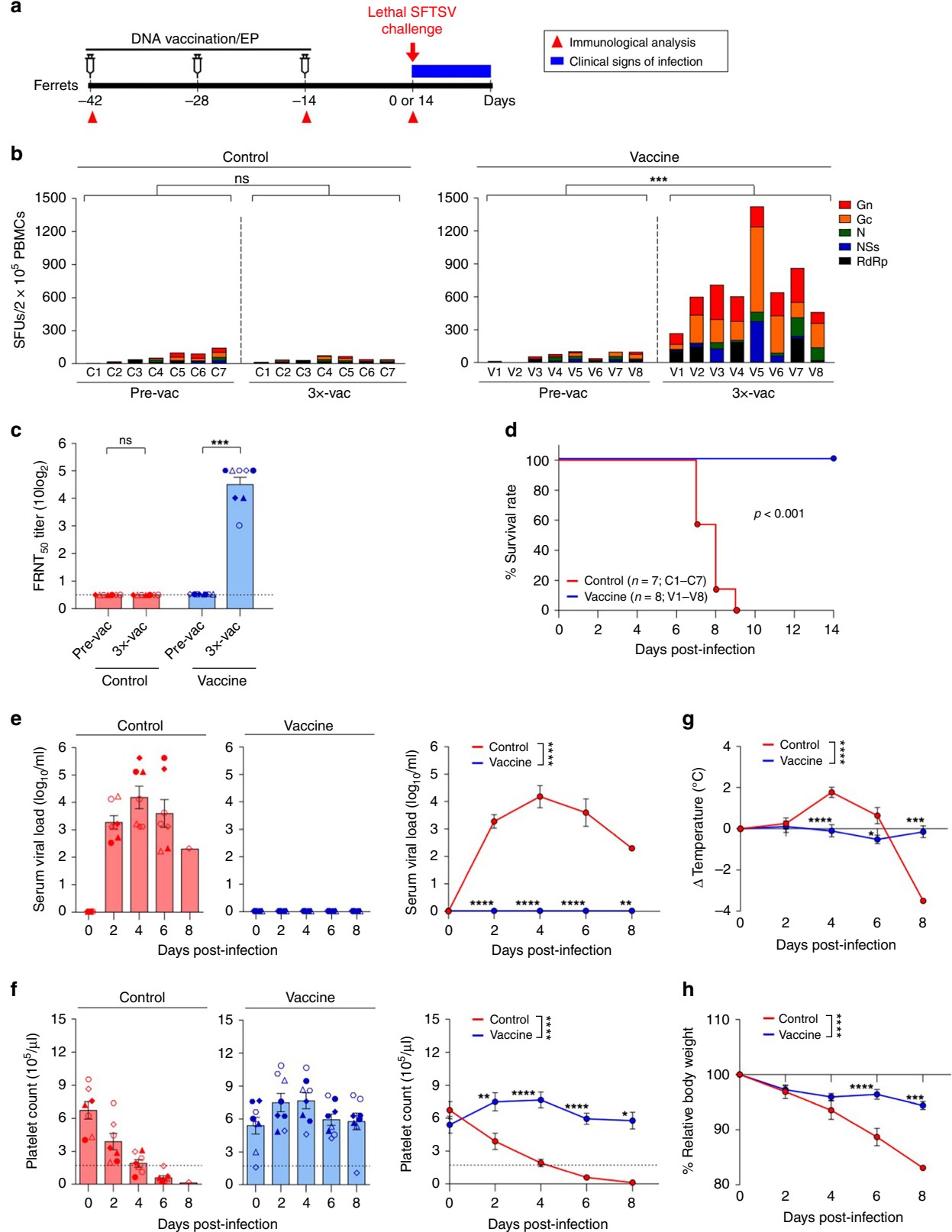

assay, respectively (Supplementary Fig. 1b). As shown in Fig. 2b, DNA vaccinations elicited robust SFTSV-specific T-cell responses in all eight ferrets. When we examined the neutralizing antibody responses to SFTSV CB1/2014 strain (genotype B), all vaccinated ferrets induced strong neutralization activity (Fig. 2c).

To evaluate the protective efficacy of our SFTSV DNA vaccines, vaccinated and non-vaccinated (control) ferrets were intramuscularly challenged with a lethal-dose of SFTSV ($10^{7.6}$ $TCID_{50}$ of the SFTSV CB1/2014 strain (genotype B)) 2 weeks (C1–C4 and V1–V4) or 4 weeks (C5–C7 and V5–V8) after the

**Fig. 2** SFTSV vaccines confer complete protection against lethal SFTSV challenge in ferrets. **a** Immunization schedule (Supplementary Fig. 1b presents a detailed timeline of vaccination and immune analysis). **b** T-cell immunogenicity of SFTSV DNA vaccines in ferrets. IFN-γ ELISpot assays were performed to detect antigen-specific IFN-γ-producing T cells by stimulating peripheral blood mononuclear cells (PBMCs) from vaccinated or non-vaccinated (control) ferrets with OLP pools. Data represent the average number of spot-forming units (SFUs) per $2 \times 10^5$ PBMCs. **c** Neutralizing antibody response to SFTSV CB1/2014 strain (genotype B) generated by SFTSV DNA vaccines in vaccinated ferrets. The amount of neutralizing antibody against SFTSV was determined based on FRNT$_{50}$. **d** Survival of vaccinated ($n = 8$, blue line) and non-vaccinated control ($n = 7$, red line) ferrets after lethal SFTSV challenge. **e** Circulating viral titers of vaccinated (blue bars, symbols, and line) and non-vaccinated (red bars, symbols and line) ferrets after SFTSV challenge. Blood was collected from ferrets every other day, and viral copy numbers were determined by real-time PCR. **f** Platelet counts of vaccinated (blue bars, symbols and line) and non-vaccinated (red bars, symbols, and line) ferrets after SFTSV challenge. The normal platelet count range in ferrets is $171.7$–$1280.6 \times 10^3$ per μL. Dashed lines indicate normal platelet count values. **g**, **h** Temperature (**g**) and temperature relative weight (**h**) of vaccinated (blue line) and non-vaccinated (red line) ferrets after SFTSV challenge. Data are presented as mean ± s.e.m. Asterisks next to the symbols on graphs indicate significance between vaccinated and control ferrets per dpi. Asterisks next to the group labels indicate significance between groups indicated. Statistical significance was determined by two-tailed Mann–Whitney *U*-test (**b**, **c**), log-rank (Mantel-Cox) test (**d**), or two-way ANOVA test with Sidak correction (**e–h**). *$p < 0.05$; **$p < 0.01$; ***$p < 0.001$; ****$p < 0.0001$. Source data are provided as a Source Data file

third vaccination. Amino acid sequence homology between DNA vaccine and challenge SFTSV strain were 99.4% (Gn), 98.1% (Gc), 99.2% (N), 98.6 (NSs), and 99.6% (RdRp). After the challenge, all ferrets were monitored for clinical signs of infection, which included survival rate, viral titers and platelet counts in the blood, body weight, and body temperature. All vaccinated ferrets survived, whereas all control ferrets died within 10-days post infection (Fig. 2d, $p < 0.001$, Mantle-Cox test). Control ferrets manifested significant viremia post challenge. In contrast, no viremia was observed in any of the vaccinated ferrets throughout the course of SFTSV challenge, demonstrating the induction of sterilizing immunity (Fig. 2e, $p < 0.0001$, repeated-measures analysis of variance (ANOVA)). Whereas all control ferrets developed severe thrombocytopenia from day 4 after SFTSV challenge, vaccinated ferrets had no significant reduction in blood platelet or white blood cell (WBC) counts (Fig. 2f, $p < 0.0001$, repeated-measures ANOVA). In addition, vaccinated ferrets exhibited no significant changes in body temperature or body weight after challenge vs. control ferrets that manifested high fevers and 20% weight loss (Fig. 2g–h, $p < 0.0001$ and $p < 0.0001$, respectively). The ferrets challenged at 2 weeks after their last vaccination and those challenged at 4 weeks after their last vaccination did not significantly differ in terms of serum viral load, platelet counts, WBC counts, temperature changes, or % relative body weight (Supplementary Fig. 4).

Triple-vaccinated ferrets exhibited complete protection against lethal SFTSV challenge; therefore, we next examined the protection efficacy after vaccination once or twice. Six naive aged-ferrets were vaccinated with SFTSV DNA vaccines (pVax1-Gn, pVax1-Gc, pVax1-N, pVax1-NSs, and pVax1-RdRp) twice ($n = 3$, V9–V11) or once ($n = 3$, V12–V14) (Supplementary Fig. 1b). All twice-vaccinated ferrets exhibited strong SFTSV-specific T-cell responses and strong neutralizing activity (Supplementary Fig. 5a, b), and survived without developing any clinical signs of infection after challenge with a lethal-dose of SFTSV (CB1/2014 strain) (Supplementary Fig. 5c–h). However, ferrets vaccinated once (V12–V14) manifested significant viremia and high fever after SFTSV challenge (Supplementary Fig. 5d, e, g) and two out of three ferrets died after lethal challenge (Supplementary Fig. 5c). Notably, one ferret (V13) showed that a single vaccination still induced strong SFTSV-specific T-cell responses (Supplementary Fig. 5a). This ferret survived lethal SFTSV challenge and was completely cleared of virus at 14 days post challenge (Supplementary Fig. 5c, d). Collectively, these results suggest that twice DNA vaccination appears to be sufficient for complete protection against lethal SFTSV infection, and that single DNA vaccination achieves only partial protection.

We next examined the protection efficacy when using lower amounts of DNA vaccines. Three naive aged-ferrets per group

were vaccinated with a total of 200 μg DNA vaccine (40 μg of each plasmid, V15–V17) or a total of 40 μg DNA vaccines (8 μg of each plasmid, V18–V20) (Supplementary Fig. 1b). Strikingly, vaccination with only 40 μg of DNA vaccine (V18–V20) induced strong SFTSV-specific T-cell responses and strong neutralizing activity (Supplementary Fig. 6a, b). All of these vaccinated ferrets survived when challenged with a lethal-dose of SFTSV (CB1/2014 strain), without thrombocytopenia, although one of the three ferrets manifested very low viremia (1.1 $\log_{10}$ copies per mL) at 2 days post challenge (Supplementary Fig. 6c–h). The ferrets vaccinated with 200 μg of DNA vaccine (V15–V17) were completely protected from lethal SFTSV challenge, with no clinical signs of infection (Supplementary Fig. 6c–h). These results suggest that even a very small amount of DNA vaccine can induce sufficient SFTSV-specific immune responses to protect from lethal SFTSV challenge.

Overall, our data showed that SFTSV DNA vaccination with in vivo electroporation could induce sterilizing immunity, and confer complete protection against lethal SFTSV infection in ferrets.

**Anti-envelope antibodies are crucial for immune protection.** We next examined which SFTSV antigen among Gn/Gc, N, NSs, and RdRp was the most important antigen for inducing protective immunity against lethal SFTSV infection, and which types of immune responses (antibody responses or T-cell responses) were critical for immune protection in vaccinated ferrets.

In the first set of experiments to examine the protective efficacy of DNA vaccines encoding envelope proteins, four naive aged-ferrets (>4-years-old, V21–V24) were immunized with a mixture of Gn and Gc vaccine candidates three times at 2-week intervals via intradermal injection followed by in vivo electroporation (Fig. 3a and Supplementary Fig. 1c). Another six naive aged-ferrets, comprising the control group (C8–C13), were immunized with the control plasmid backbone (modified pVax1) followed by in vivo electroporation. When vaccine-induced immune responses in four ferrets were evaluated by IFN-γ ELISPOT assays and in vitro neutralization assays, we observed that all four vaccinated ferrets elicited vigorous Gn/Gc-specific T-cell responses and strong neutralization activity after the third DNA vaccination (Fig. 3b, c). After lethal SFTSV challenge of vaccinated ferrets, all Gn/Gc DNA-vaccinated ferrets survived, whereas all control ferrets died within 10 days of challenge (Fig. 3d, $p < 0.005$ (Gn/Gc vaccine vs. control, Mantle-Cox test)). In contrast to the control ferrets, 3 of 4 Gn/Gc-vaccinated ferrets had no detectable viremia throughout the course of challenge (one vaccinated ferret exhibited very low-level viremia at day-2 and then completely cleared the virus at subsequent timepoints) and all vaccinated ferrets maintained normal platelet counts after

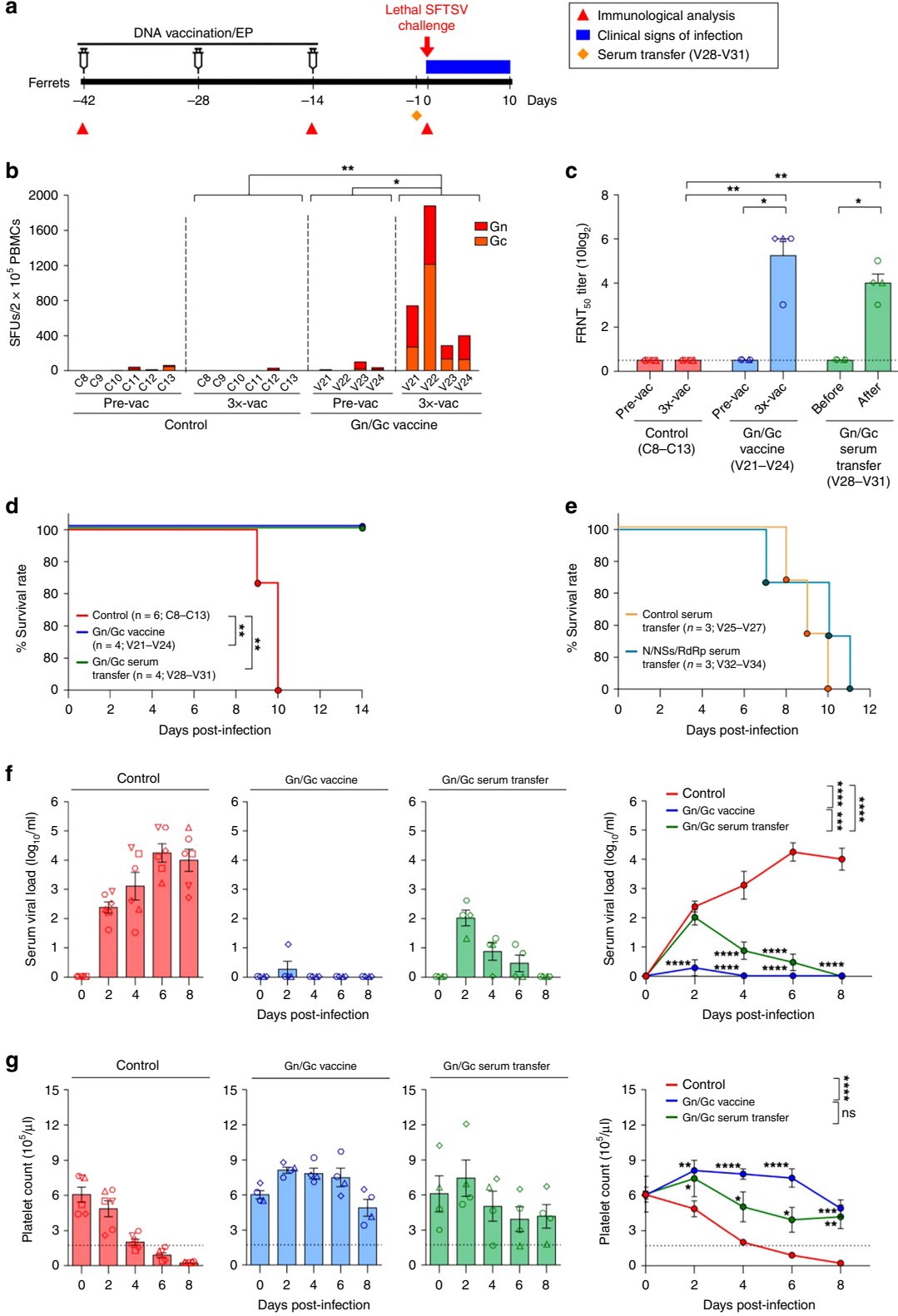

SFTSV challenge (Fig. 3f–g, blue bars (symbols) and lines, $p < 0.0001$ and $p < 0.0001$, respectively (Gn/Gc vaccine vs. control, repeated-measures ANOVA)). Moreover, these vaccinated ferrets showed no other clinical signs of infection, including WBC counts, body temperature, body weight, and ALT/AST levels (Supplementary Fig. 7). These results suggest that Gn- and Gc-specific immune responses played a critical role in protection against lethal SFTSV infection.

To specifically investigate the role of the anti-envelope antibody response as a protective immunity against SFTSV infection, sera from Gn/Gc-vaccinated ferrets (S3-S4, Supplementary Fig. 8) were collected after three rounds of vaccination

**Fig. 3** Anti-envelope antibodies can protect ferrets against lethal SFTSV challenge. **a** Immunization schedule (Supplementary Fig. 1c shows a detailed timeline of vaccination and immune analysis). **b** T-cell immunogenicity of SFTSV DNA vaccines in ferrets. IFN-γ ELISpot assays were performed to detect antigen-specific IFN-γ-producing T cells by stimulating PBMCs from vaccinated or non-vaccinated (control) ferrets with OLP pools covering Gn and Gc. Data represent the average number of SFUs per $2 \times 10^5$ PBMCs. **c** Neutralizing antibody response to SFTSV CB1/2014 strain (genotype B) generated in ferrets by DNA vaccines (pVax1-Gn and pVax1-Gc, blue bars and symbols) or serum transfer (green bars and symbols). The amount of neutralizing antibody against SFTSV was determined based on $FRNT_{50}$. **d** Survival of non-vaccinated control ($n = 6$, red line), Gn/Gc-vaccinated ($n = 4$, blue line), and Gn/Gc serum-transferred ($n = 4$, green line) ferrets after lethal SFTSV challenge. **e** Survival of control serum-transferred ($n = 3$, orange line) and N/NSs/RdRp serum-transferred ($n = 3$, light blue line) ferrets after lethal SFTSV challenge. **f** Circulating viral titers in non-vaccinated control (red bars, symbols and line), Gn/Gc-vaccinated (blue bars, symbols, and line), and Gn/Gc serum-transferred (green bars, symbols, and line) ferrets after SFTSV challenge. **g** Platelet counts in non-vaccinated control (red bars, symbols and line), Gn/Gc-vaccinated (blue bars, symbols, and line), and Gn/Gc serum-transferred (green bars, symbols and line) ferrets after SFTSV challenge. The normal platelet count range in ferrets is $171.7-1280.6 \times 10^3$ per μL. Dashed lines indicate the normal platelet count values. Data are presented as mean ± s.e.m. Asterisks next to the symbols on graphs indicate significance between vaccinated and control ferrets per dpi. Asterisks next to the group labels indicate significance between groups indicated. Statistical significance was determined by two-tailed Mann–Whitney $U$-test (**b**, **c**), log-rank (Mantel–Cox) test (**d**, **e**), or two-way ANOVA test with Sidak correction (**f**, **g**). *$p < 0.05$; **$p < 0.01$; ***$p < 0.001$; ****$p < 0.0001$. Source data are provided as a Source Data file

and transferred into four naive aged-ferrets (7 mL serum per ferret, V28–V31) via intraperitoneal injection 1 day before SFTSV challenge (Supplementary Fig. 1c). All ferrets that received sera from Gn/Gc-vaccinated ferrets had reciprocal $FRNT_{50}$ neutralizing antibody titers of 80–320 after serum transfer (Fig. 3c, green bars and symbols), which were comparable to those in Gn/Gc-vaccinated ferrets. When these ferrets were challenged with a lethal-dose of SFTSV (CB1/2014 strain), all ferrets that received serum from Gn/Gc-vaccinated ferrets survived, whereas all control ferrets died within 10 days of challenge (Fig. 3d, $p < 0.01$ (vs. control, Mantle-Cox test)). All Gn/Gc serum-transferred ferrets exhibited low-level viremia in blood at 2 days after challenge (ranged from 1.3 to 2.6 $\log_{10}$ copies per mL), but completely cleared virus thereafter (Fig. 3f, green bars (symbols) and green lines, $p < 0.0001$ (vs. control, repeated-measures ANOVA). Moreover, none of the Gn/Gc serum-transferred ferrets exhibited severe thrombocytopenia (Fig. 3g, green bars (symbols) and green lines) or other significant clinical symptoms of SFTS (Supplementary Fig. 7) following SFTSV challenge.

As control groups, we investigated the protection efficacy of the transfer of serum from pVax1(empty vector)-vaccinated ferrets ($n = 3$, 7 mL serum per ferret, V25–V27) or serum from N/NSs/RdRp-vaccinated ferrets ($n = 3$, 7 mL serum per ferret, V32–V34). No ferret that received serum from pVax1-vaccinated ferrets or N/NSs/RdRp-vaccinated ferrets exhibited any detectable neutralization activity after passive serum transfer (Supplementary Fig. 8 and Supplementary Fig. 9a). All ferrets that were passively transferred with serum from pVax1-vaccinated or N/NSs/RdRp-vaccinated ferrets died within 11 days following lethal SFTSV challenge, after exhibiting typical clinical signs of SFTS, including high viral titers, severe thrombocytopenia, high fever, and body weight loss (Fig. 3e and Supplementary Fig. 9b–h).

Collectively, these results suggest that anti-Gn and -Gc antibodies are sufficient for protective immunity against SFTSV infection, and that non-neutralizing antibodies specific to non-envelope proteins do not play a role in this protective immunity.

**Non-envelope-specific T cells can confer immune protection.** Next, we sought to investigate the protective efficacy of the non-envelope (N, NSs, and RdRp) vaccines. Four naive aged-ferrets (V35–V38) were immunized with a mixture of the N, NSs, and RdRp vaccine candidates (pVax1-N, pVax1-NSs, and pVax1-RdRp) (Supplementary Fig. 1c). As shown in Fig. 4a, N/NSs/RdRp DNA vaccination elicited vigorous T-cell responses to desired antigens in all four vaccinated ferrets. However, in vitro neutralization assays revealed that no N/NSs/RdRp-vaccinated ferrets had any detectable neutralization activity after

vaccinations (Fig. 4b), indicating that only Gn/Gc DNA vaccines, not N/NSs/RdRp DNA vaccines, can induce anti-SFTSV neutralizing antibody responses. When these N/NSs/RdRp-vaccinated ferrets (V35–V38) were challenged with a lethal-dose of SFTSV (CB1/2014 strain), all vaccinated ferrets survived (Fig. 4c, $p < 0.01$ (vs. control), Mantle-Cox test)). All vaccinated ferrets exhibited low-level viremia in blood at 2 days after challenge (ranged from 1.3 to 2.1 $\log_{10}$ copies per mL), but successfully cleared virus thereafter (Fig. 4d, green bars (symbols) and lines, $p < 0.0001$ (vs. control, repeated-measures ANOVA)). Additionally, after SFTSV challenge, all vaccinated ferrets maintained normal platelet counts (Fig. 4e, $p < 0.0005$ vs. control, repeated-measures ANOVA) and exhibited no significant clinical symptoms of SFTS in terms of WBC counts, body temperature, body weight, and ALT/AST levels (Supplementary Fig. 10). Taken together, these results suggest that DNA vaccine-induced T-cell responses specific to non-envelope proteins (N, NSs, and RdRp) also can confer immune protection against lethal SFTSV infection.

Next, we investigated the protection efficacy of individual DNA vaccines encoding N, NS, or RdRp to determine which antigen among the SFTSV non-envelope proteins is most important for inducing protective immunity. Three naive aged-ferrets in each group were immunized with the N (V39–V41), NSs (V42–V44), or RdRp (V45–V47) DNA vaccine. After vaccination, all ferrets exhibited significant T-cell responses to desired SFTSV antigen with no detectable neutralization activity (Fig. 4a, b). After lethal SFTSV challenge, all individual DNA-vaccinated nine ferrets (V39–V47) manifested a significantly higher viral load (Fig. 4d) with clinical symptoms (mild thrombocytopenia, high fever, and slight body weight loss) (Fig. 4e and Supplementary Fig. 10), compared to N/NSs/RdRp-vaccinated ferrets (V35–V38). Notably, one N DNA-vaccinated ferret (V39), one NSs DNA-vaccinated ferret (V42), and one RdRp DNA-vaccinated ferret (V45) survived after SFTSV challenge with complete viral clearance and normal platelet/WBC counts (Fig. 4c–e and Supplementary Fig. 10a). Our results suggest that individual DNA vaccines containing N, NSs, or RdRp can contribute to only partial, not complete, protection against SFTSV infection.

In conclusion, we developed a DNA vaccine for SFTSV and evaluated its immunogenicity and protective efficacy using a lethal infection ferret model. In particular, we found that ferrets immunized with DNA vaccines encoding all SFTSV proteins were completely protected from lethal SFTSV challenge without developing any clinical signs, demonstrating the induction of sterilizing immunity. This suggests that DNA vaccine platform might represent a suitable vaccine strategy against SFTSV due to its ability to induce a wider range of immune response types[23,24]. Although it was recently reported that a live attenuated

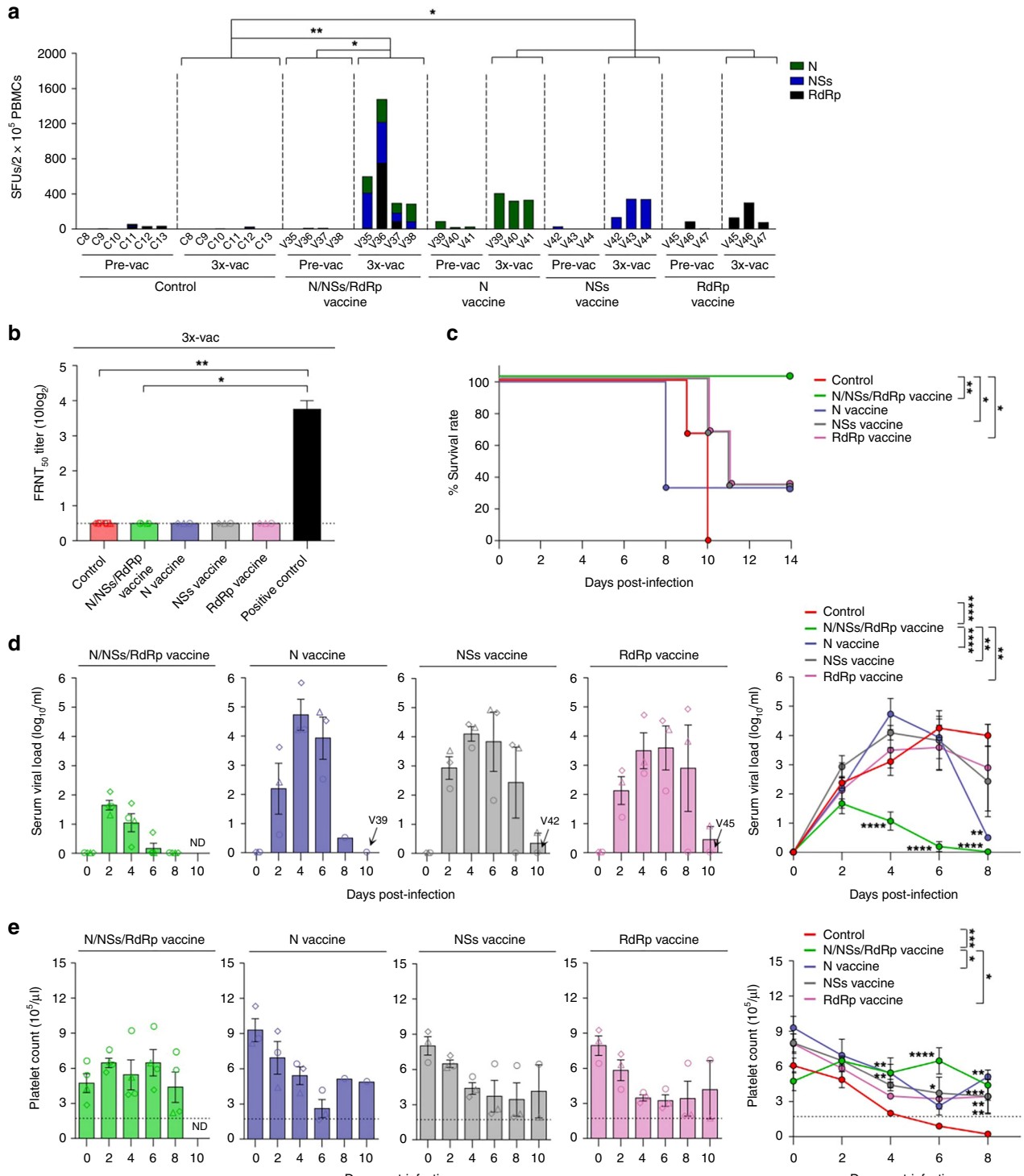

**Fig. 4** N/NSs/RdRp-specific T-cell responses can protect ferrets against lethal challenge. **a** T-cell immunogenicity of SFTSV DNA vaccines in ferrets. IFN-γ ELISpot assays were performed to detect antigen-specific IFN-γ-producing T cells by stimulating PBMCs from vaccinated or non-vaccinated (control) ferrets with OLP pools covering N, NSs, and RdRp. Data represent the average number of SFUs per $2 \times 10^5$ PBMCs. **b** Neutralizing antibody response to SFTSV CB1/2014 strain (genotype B) generated by DNA vaccines in ferrets. The amount of neutralizing antibody against SFTSV was determined based on $FRNT_{50}$. **c** Survival of N/NSs/RdRp- ($n = 4$, green line), N- ($n = 3$, indigo line), NSs- ($n = 3$, gray line), or RdRp- ($n = 3$, pink line) vaccinated and non-vaccinated control ($n = 6$, red line) ferrets after lethal SFTSV challenge. **d** Circulating viral titers of N/NSs/RdRp- (green bars, symbols and line), N- (indigo bars, symbols, and line), NSs- (gray bars, symbols and line), or RdRp- (pink bars, symbols and line) vaccinated ferrets after SFTSV challenge. **e** Platelet counts in N/NSs/RdRp- (green bars, symbols, and line), N- (indigo bars, symbols, and line), NSs- (gray bars, symbols, and line), or RdRp- (pink bars, symbols, and line) vaccinated ferrets after SFTSV challenge. The normal platelet count range in ferrets is $171.7-1280.6 \times 10^3$ per μL. Dashed lines indicate normal platelet count values. Data are presented as mean ± s.e.m. Asterisks next to the symbols on graphs indicate significance between vaccinated and control ferrets per dpi. Asterisks next to the group labels indicate significance between groups indicated. ND, not determined due to the sample unavailability. Statistical significance was determined by two-tailed Mann–Whitney U-test (**a**, **b**), log-rank (Mantel-Cox) test (**c**), two-way ANOVA test with Sidak correction (**d**, **e**). *$p < 0.05$; **$p < 0.01$; ***$p < 0.001$; ****$p < 0.0001$. Source data are provided as a Source Data file

recombinant vesicular stomatitis virus (rVSV)-based vaccine expressing the SFTSV Gn/Gc glycoproteins can elicit protection against SFTSV in immunocompromised IFNAR$^{-/-}$ mice[25], our current study represent a substantial contribution towards the development of an effective preventive vaccine for SFTS, considering that our SFTSV DNA vaccines induced complete protection against lethal SFTSV challenge in an immunocompetent middle-sized animal model that exhibits clinical manifestations seen in SFTS patients[18]. Moreover, in the present study, we investigated the most effective SFTSV antigens for inducing protective immunity against lethal SFTSV infection. Our results indicated that Gn/Gc may be the most effective antigens for inducing protective immunity. Moreover, DNA vaccines encoding SFTSV N, NSs, or RdRp may provide an additional protective effect when used in combination with Gn/Gc DNA vaccines as we showed that the inoculation of Gn/Gc DNA vaccines together with non-envelope DNA vaccines induced sterilizing immunity in all vaccinated ferrets. Few prior studies have examined the characteristics of protective immunity in response to SFTSV infection in terms of the type of vaccine-induced immunity and the essential vaccine antigens; thus, our present study provides valuable insights into the design of preventive vaccines for SFTS.

## Methods

**DNA plasmids.** Five different SFTSV DNA plasmids were generated using a modified pVax1 mammalian expression vector and genes encoding Gn, Gc, N, NSs, and RdRp based on the common sequences of 31 SFTSV strains isolated from South Korea, China, and Japan. The genes encoding Gn, Gc, N, NSs, and RdRp were genetically optimized for enhanced expression, including codon and RNA optimization and then subcloned into modified pVax1 under control of the cytomegalovirus immediate-early promoter. A Kozak sequence and immunoglobulin E leader sequence were added to facilitate expression. Each construct was synthesized and the sequences verified by GeneArt (Germany). The five SFTSV DNA plasmids were designated pVax1-Gn, pVax1-Gc, pVax1-N, pVax1-NSs, and pVax1-RdRp (Fig. 1a).

**Western blot.** 293T cells (ATCC, CRL-3216) were transfected with 12 μg of pVax1-Gn, pVax1-Gc, pVax1-N, pVax1-NSs, or pVax1-RdRp. The cells were incubated for 48 h, and then harvested using RIPA buffer (Thermo Fisher Scientific). Each cell lysate was loaded onto sodium dodecyl sulfate–polyacrylamide gel electrophoresis (SDS/PAGE) gels, and then the separated proteins were transferred onto a membrane. The membrane was incubated overnight at 4 °C with pVax1-Gn, pVax1-Gc, pVax1-N, pVax1-NSs, or pVax1-RdRp-vaccinated mouse serum (diluted 1:300) and rabbit anti-β-actin antibody (Abcam, ab8227, 1:5000). The signal was detected using horseradish peroxidase-conjugated secondary antibody (Abcam, ab97023, 1:100,000; Abcam, ab97051, 1:100,000) with SuperSignal West Femto Maximum Sensitivity Substrate (Thermo Fisher Scientific). Uncropped and unprocessed scans of the western blots are provided as a Source Data file.

**Ethical statement.** For all experiments using mice and ferrets, we have complied with all relevant ethical regulations for animal testing and research. Mouse care and experimental procedures were performed with the approval from the Animal Care Committee of Korea Advanced Institute of Science and Technology. All ferret experiments were approved by the Medical Research Institute, a member of Laboratory Animal Research Center of Chungbuk National University (LARC) (approval number: CBNUA-986-16-01) and were conducted in strict accordance and adherence to relevant policies regarding animal handling as mandated under the Guidelines for Animal Use and Care of the Korea Centers for Disease Control and Prevention (KCDC). Viruses were handled in an enhanced biosafety level 3 (BSL3) containment laboratory as approved by the Korea Centers for Disease Control and Prevention (KCDC-14-3-07).

**Animals and immunization.** For vaccination studies in mice, 5 to 6-week-old female BALB/c mice were purchased from DBL (Republic of Korea). All mice were maintained under conventional conditions. BALB/c mice were immunized into the internal thigh muscle twice at 3-week intervals with a mixture of 40 μg each pVax1-Gn, pVax1-Gc, pVax1-N, pVax1-NSs, and pVax1-RdRp in 120 μL of phosphate-buffered saline (PBS), followed by in vivo electroporation using a CELLECTRA®-3P device (Inovio Pharmaceuticals, Inc.). Control mice were immunized with 200 μg of pVax1 vector in 120 μL of PBS. Three weeks after the last immunization, the mice were sacrificed for immunological examination of the spleen and the blood. Detailed experimental strategy is presented in Supplementary Fig. 1a.

For vaccination studies in ferrets, outbred ferrets (*Mustela putorius furo*) > 4 years of age were purchased from Idbio (Republic of Korea). The naive aged-ferrets

were intradermally immunized three times at 2-week intervals with 200 μg each of pVax1-Gn, pVax1-Gc, pVax1-N, pVax1-NSs, and/or pVax1-RdRp in 200 μL of PBS, which was followed by in vivo electroporation using a CELLECTRA®-3P device (Inovio Pharmaceuticals, Inc.) on the internal thigh. Control naive ferrets were immunized with 1 mg of pVax1 vector in 200 μL of PBS. Peripheral blood samples were collected on days 0, 28, and 42 post immunization for immunological examination. Detailed experimental strategies are presented in Supplementary Fig. 1b, c. In ferret studies, DNA vaccines were administered via an intradermal route rather than intramuscularly. Studies in NHPs, rabbits, and humans show that intradermal delivery of DNA vaccine yields equal or greater immunogenicity and protection[26–29]. Compared to mice, ferrets have a thicker epidermal and dermal layer, allowing for intradermal delivery.

**Passive transfer of serum from vaccinated to naive ferrets.** Seven-milliliter serum from pVax1-Gn/pVax1-Gc-vaccinated ferrets was administered intraperitoneally to aged-naive ferrets 1 day prior to SFTSV challenge (Supplementary Figs. 1c and 8). As control groups, 7 mL serum from N/NSs/RdRp-vaccinated ferrets or pVax1-vaccinated ferrets was administered intraperitoneally to aged-naive ferrets 1 day prior to SFTSV challenge (Supplementary Figs. 1c and 8). Neutralizing antibody titers in the blood of passively immunized ferrets were evaluated by FRNT$_{50}$ at the time of SFTSV challenge.

**Overlapping peptides.** OLPs were synthesized as 15-mers that overlapped by eight amino acids to cover the whole amino acid sequence of Gn, Gc, N, NSs, and RdRp of DNA vaccines (Mimotopes). Lyophilized peptides were solubilized in 5% dimethyl sulfoxide (DMSO; Sigma-Aldrich). The concentration of each peptide in the pools was 25 μg mL$^{-1}$, which was finally diluted to 1 μg mL$^{-1}$ for stimulation of splenocytes and peripheral blood mononuclear cells (PBMCs) to examine vaccine-induced SFTSV-specific T-cell responses in mice and ferrets, respectively.

**IFN-γ ELISpot assays.** IFN-γ ELISpot assays were performed to measure antigen-specific IFN-γ secretion[30]. One-hundred microliter of anti-mouse IFN-γ antibody (clone AN-18, eBioscience, 16-7313-81, 2 μg mL$^{-1}$) were added onto hydrophobic polyvinylidene difluoride (PVDF) plates (Multiscreen, Millipore). Plates were incubated overnight at 4 °C. Plates were washed away with PBS and blocked with 1% bovine serum albumin (BSA; Bovogen) for 1 h at room temperature (RT). The spleens of immunized mice were collected in RPMI–1640 medium (WelGENE, Daegu, Republic of Korea) supplemented with 5% fetal bovine serum (FBS) and 1x penicillin–streptomycin, mechanically mashed, and filtered using 40 μm strainers. After centrifugation, cells were treated with RBC lysis buffer (BioLegend, San Diego, CA, USA) for 5 min at room temperature, washed, and then resuspended ($0.5 \times 10^6$ per well) in RPMI medium. Splenocytes were stimulated with SFTSV OLP pools (1 μg per mL of each peptide) for 24 h at 37 °C in a 5% CO$_2$ atmosphere. PMA (10 ng mL$^{-1}$, Sigma-Aldrich) and ionomycin (500 ng mL$^{-1}$, Sigma-Aldrich) were used as a positive control and 5% DMSO (Sigma-Aldrich) served as a negative control. Plates were washed away with PBS and 0.05 % Tween/PBS. One-hundred microliters of the biotinylated anti-mouse biotinylated IFN-γ mAbs (clone R4-6A2, eBioscience, 0.5 μg mL$^{-1}$) were added into each well and incubated for 1 h at RT. Thereafter, plates were incubated for 1 h at RT with 100 μL of streptavidin-alkaline phosphatase (Invitrogen, 1:5,000). One-hundred microliters of BCIP/NBT substrate solution (Bio-Rad) were added into each well and incubated for 10 min at RT. The development was stopped by washings with tap-water. The ferret IFN-γ ELISpot Plus assay (Mabtech) was performed as recommended by the manufacturer. PBMCs were isolated from whole blood by Ficoll-Hypaque density gradient centrifugation and resuspended ($0.2 \times 10^6$ per well) in RPMI medium. The PBMCs were stimulated with SFTSV OLP pools for 40 h at 37 °C in a 5% CO$_2$ atmosphere. Spots were enumerated using an automated ELISpot reader (AID GmbH,) and the number of specific spots was calculated by subtracting the number of spots in negative control wells from the number of spots in OLP pool-stimulated wells.

**Intracellular cytokine staining assay.** Mouse splenocytes were stimulated ex vivo with SFTSV OLP pools. Brefeldin A (GolgiPlug, BD Biosciences) and monensin (GolgiStop, BD Biosciences) were added to the culture 1 h after stimulation and the culture was maintained for 5 h. The Live/Dead Fixable Red Dead Cell stain kit (Invitrogen), anti-CD19–PE-CF594 (clone 1D3, BD Biosciences, 562291, 1:100), anti-CD3–BV510 (clone 145-2C11, BD Biosciences, 563024, 1:100), anti-CD4–Alexa Flour 700 (clone RM4-5, BD Biosciences, 557956, 1:100), and anti-CD8–APC-H7 (clone 53-6.7, BD Biosciences, 560182, 1:100) were used for immunostaining at 4 °C for 20 min. For intracellular staining, cells were permeabilized and fixed using the BD Cytofix/Cytoperm kit (BD Biosciences) at 4 °C for 20 min, which was followed by staining with anti-IFN-γ–APC (clone XMG1.2, BioLegend, 505809, 1:100), anti-TNF–PE (clone MP6-XT22, BD Biosciences, 554419, 1:100), and anti-IL-2–PE-Cy7 (clone JES6-5H4, BD Biosciences, 560538, 1:100) at 4 °C for 20 min. All data were collected using a LSRII flow cytometer (BD Biosciences), and Boolean gating was performed using FlowJo software to analyze the polyfunctionality of T cells. Charts were visualized using SPICE software (https://niaid.github.io/spice/).

**In vitro neutralization assay**. Neutralizing antibody against SFTSV in anti-sera was determined based on FRNT$_{50}$[31]. To determined FRNT50 values of immunized ferret sera, tenfold serially diluted sera were mixed with 100 focus-forming units of SFTSV CB1/2014 strain (genotype B) and incubated for 1 h. The mixture was then used to inoculate confluent Vero E6 cells prepared in 96-well plates, and then incubated for 1 h at 37 °C. The inoculums were then removed and replaced with Dulbecco's Modified Eagle Medium (DMEM) containing 1% FBS and the cells incubated for an additional 7 days, at which time the cells were fixed with 10% neutral-buffered formalin and then blocked with 3% BSA and treated with 10% triton X-100. The cells were then incubated with mouse anti-N antibodies followed by HRP-mouse antibodies (The Jackson ImmunoResearch Laboratory, 115-035-146, 1:2,000). The visualization of foci formation in SFTSV-infected cells was performed using the 3,3′-diaminobenzidine (DAB) substrate kit (Vector Laboratories). FRNT$_{50}$ values were determined as a reciprocal of the highest dilution at which the number of foci was < 50% of the number obtained without serum.

**Identification of H-2L$^d$-restricted minimal epitope**. T-cell epitopes were mapped within NSs antigen. IFN-γ ELISpot assays were performed with splenocytes from vaccinated mice by stimulation with a single peptide (15-mer; total 41 peptides). Peptides with high numbers of IFN-γ spots were identified, and the epitope sequence corresponding to high-affinity binding to mouse MHC class I, H-2L$^d$ was identified using NetMHC 4.0 (Technical University of Denmark). The epitope peptide YPYLMAHYL was used to synthesize the H-2L$^d$ dextramer (H-2L$^d$ NSs$_{247-255}$ dextramer, Immunedex). Splenocytes of vaccinated mice were stained with PE–conjugated H-2L$^d$ NSs$_{247-255}$ dextramer at 4 °C for 20 min, followed by washing with FACS buffer (PBS containing 1% FBS and 0.05% sodium azide). For live/dead and surface staining, the following reagent and antibodies were used: Live/Dead Fixable Red Dead Cell stain kit (Invitrogen), anti-CD19–PE-CF594 (clone 1D3, BD Biosciences, 562291, 1:100), anti-CD8–BV510 (clone 53-6.7, BD Biosciences, 560776, 1:100), anti-CD62L–BV605 (clone MEL-14, BD Biosciences, 563252, 1:50), anti-CD44–BV650 (clone IM7, BioLegend, 103049, 1:50), and anti-CD3–Alexa Fluor 700 (clone 500A2, eBioscience, 56-0033-82, 1:100).

**Lethal challenge of SFTSV in ferrets**. Vaccinated or control ferrets were challenged intramuscularly with 10$^{7.6}$ TCID$_{50}$ of the SFTSV CB1/2014 strain (0.5 mL in the outside of each thigh of both legs). Throughout the ferret challenge studies, we used a TCID$_{50}$ dose (~10$^{7.6}$) of SFTSV that caused 100% fatal infection in ferrets, as in our previous recent study[18]. The survival of challenged ferrets was monitored for 14 days after lethal SFTSV challenge because our recently published study demonstrated that all aged-ferrets died within 10 days post-SFTSV challenge[18]. Clinical signs of infection (viral load, platelet counts, WBC counts, ALT/AST levels, body weight, and body temperature) were monitored at 0, 2, 4, 6, and 8 days post challenge. However, if vaccinated ferrets showed any clinical signs of infection (based on platelet and WBC counts, body temperature, and/or body weight possibly due to partial protection by vaccination), they were monitored for their clinical symptoms for 14 days post challenge. Sera were collected from each ferret at 2-day intervals after infection and peripheral virus titers were determined. Hematological parameters were analyzed using EDTA-treated whole-blood samples from infected animals using the Celltac hematology analyzer (MEK-6550J/K, Nihon Kohden). Biochemical parameters of serum from infected animals were determined using Celltac α (MEK-6550, Nihon Kohden).

**Quantification of viral copy numbers by quantitative reverse transcription PCR (qRT-PCR)**. Total RNA was extracted using TRIzol LS Reagent (Thermo Fisher Scientific), and complementary DNA was generated by reverse transcription with specific (SFTSV-S primer: 5′ cccacttggacatgtgct 3′, 0.5 μM) and random primers (Takara, #3801, 0.5 μM) using the Omniscript RT Kit (Qiagen). Viral copy numbers were determined by quantitative real-time RT-PCR using an S segment-based SFTSV-specific primer set: forward primer (SFTSV-S-F) 5′ gcagttggaatcaggga 3′ and reverse primer (SFTSV-S-R) 5′ cccacttggacatgtgct 3′. Copy numbers were calculated as a ratio based on the standard control[32]. Real-time PCR reactions were performed using a SYBR Green Supermix (Bio-Rad) and a CFX96 Touch Real-Time PCR Detection System (Bio-Rad).

**Statistical analysis**. Statistical analyses were performed using Prism software 7.0 (GraphPad Software Inc., La Jolla, CA, USA). Two-tailed Mann–Whitney $U$-test, log-rank (Mantel-Cox) test, one-way ANOVA test with Tukey correction, or two-way ANOVA test with Sidak correction was used. $p < 0.05$ was considered significant.

**Reporting summary**. Further information on research design is available in the Nature Research Reporting Summary linked to this article.

## Data availability
The data that support the findings of this study are available from the corresponding author S.H.P. upon reasonable request. The source data underlying Figs. 1b, 1d–f, 2b–h, 3b–g and 4a–e and Supplementary Figs 2a–b, 2d, 3b, 4a–e, 5a–h, 6a–h, 7a–e, 8, 9a–h and 10a–e are provided as a Source Data file.

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

## Acknowledgements

This work was supported by grants from the Korea Health Technology R&D Project through the Korea Health Industry Development Institute (KHIDI), funded by the Ministry of Health & Welfare, Republic of Korea (HI15C2888 to S.H.P. and HI15C2817 to Y.K.C. and S.H.P.) and a grant from Government-wide R&D Fund project for infectious disease research, Republic of Korea (HG18C0029 to Y.K.C.). This work was also supported by a Global Ph.D. Fellowship from the National Research Foundation of Korea (NRF-2017H1A2A1046321 to J.E.K.).

## Author contributions

E.C.S., Y.K.C. and S.H.P. designed the experiments and managed the study. J.E.K., Y.I.K., S.J.P., M.A.Y., H.I.K., S.E., T.S.K., J.S., W.S.C., J.H.J., H.L., Y.C., J.A.K., M.J., Y.E.K., H.J., K.K.K., M.S.S., Y.K.C. and S.H.P. performed the experiments. J.E.K., Y.I.K., S.J.P., J.N.M., E.C.S., M.S.S., J.U.J., Y.K.C. and S.H.P. analyzed the data. J.E.K., E.C.S., J.U.J., Y.K.C. and S.H.P. conceived the work and wrote the manuscript, which was revised and approved by all authors.

## Additional information

**Competing interests:** H.L., Y.C., J.A.K., M.J., and J.N.M. are employees of GeneOne Life Science, Inc. The other authors declare no competing interests.

