## [Peer Review File · Nature Communications]

Reviewers' Comments:

Reviewer #1:

Remarks to the Author:

Kwak et. al demonstrating that using SFTSV-specific DNA vaccine can protect animals which challenged by the SFTSV. Moreover, the authors investigate the immune response activated by this DNA- vaccine. In general, this is well written MS describing the efficacy of DNA vaccine for STFSV. Overall, the approach used in this MS is not novel and the data are not surprising. However, it should be highlighted that data are solid, but the results is of interest for small scientific community (experts in the STFSV and DNA vaccine community) and lack the wider field.

Minor comments:

- It is interesting to investigate the protection efficacy by just using N and or Ns
- It would also interesting to see the protection efficacy corresponding to the different amount of Plasmid used (the lowest amount that can give protection).
- How many days post challenge he survived animal followed?
- How the Viral titers determined in figure2 and 3? Based on PCR or infectious particles?

Reviewer #2:

Remarks to the Author:

In this manuscript, the authors generate a DNA vaccine against Severe Fever with Thrombocytopenia Syndrome Virus (SFTSV) in a recently-established lethal, aged ferret model which recapitulates the clinical manifestations seen in SFTS patients. Following three vaccinations with a mixture of five DNA vaccines (against Gn, Gc, NP, RdRp, and NSs) at 2-week intervals, ferrets were lethally challenged with SFTSV. DNA vaccination conferred complete protection in the lethal, aged ferret model, while control ferrets succumbed to SFTSV infection. The authors further demonstrate the Gn/Gc are important for neutralizing antibody response in the ferret model, while the NSs/NP/RdRp elicit a T-cell specific response. While these studies show protection from lethal SFTSV infection in a novel animal model, there are major concerns that should be considered for this manuscript.

Major Concerns

While the authors present a platform for an SFTSV vaccine, a study was recently published using a single-dose of an rVSV-SFTSV/AH12-GP that protects IFNAR-/- mice from lethal SFTSV infection (Dong F, et al. npj Vaccines. 2019. doi: 10.1038/s41541-018-0096-y). The authors of the present study state that their "vaccine is the first to demonstrate complete protection against lethal SFTSV challenge using an immunocompetent, middle-size animal model with clinical manifestations of SFTSV infection", which may be misleading to the reader. The authors should refer to the previous related study and tone down the sentence above. Furthermore, while the DNA vaccine does elicit protective immunity in the animal model presented, this is only following three vaccinations, which may not be realistic in a clinical setting. Have the authors observed protection or partial protection in ferrets vaccinated once?

The authors previously established a lethal immunocompetent animal model for the study of SFTSV using outbred, aged ferrets, in which aged ferrets lethally infected with SFTSV showed clinical signs of SFTS infection: thrombocytopenia, leukopenia, fever, and elevated AST/ALT. The current manuscript does not show these additional clinical signs in their vaccinated animals or control animals. Can the authors provide rationale for leaving out these experiments in the present manuscript, which have been previously performed in the same animal model? This would strengthen the efficacy of their vaccine platform.

Immunization with the mixture of five different DNA vaccine plasmids is unusual and not explained in the text. The rationale of this vaccination strategy should be provided in the manuscript. This vaccination scheme does not help to define the most effective antigen to induce protective immunity in vaccinated animals. Moreover, the authors should perform the protection studies with individual DNA vaccine which encode the NSs, N, or RdRp. In general, immunization with nucleocapsid protein as an antigen elicits T cell-mediated protective immunity. Interestingly, in Figure 1d, vaccination potentially elicit T cell-mediated immune response against NSs. Therefore, to determine which antigen (N or NSs) is effective antigen to induce protective cell-mediated immunity, the authors should perform protection studies in the ferrets vaccinated with N or NSs DNA vaccine.

The authors investigated "the role of the anti-envelope antibody response as a protective immunity against SFTSV infection" by transferring sera from Gn/Gc vaccinated ferrets to naive ferrets. While this experimental outline and rationale are clear, the authors lack an experimental control for these studies; for the results presented, serum-transferred ferret survival rates, viral loads, and platelet counts are directly compared to control-vaccinated ferrets. Control vaccinated ferret serum transfer to naïve ferrets should be done in parallel with Gn/Gc vaccinated animal serum transfer to naïve animals. Furthermore, the authors did not comment on any other clinical disease signs in vaccinated or serum transfer animals (fever, leukopenia, AST/ALT) even though they had low viral titers.

Minor Concerns:

Line 59-61: According to current ICTV classification, SFTSV is a member of the genus *Bangyangvirus* in the family *Phenuiviridae* of the order *Bunyavirales*. The sentence should be corrected with correct taxonomical classification.

Line 61-62: The authors write that the "the L segment encodes the viral RNA polymerase". This should be changed to the "viral RNA-dependent RNA polymerase".

Line 63-64: The authors write "SFTSV is an arbovirus transmitted by the tick *Haemaphysalis longicornis*", however, *Rhipicephalus microplus*, and others are vectors of SFTSV (Yun, SM. *Emerg Infect Dis* 2014. doi: 10.3201/eid2008.131857). The authors should consider re-writing this sentence to include other vectors, or may want to indicate *Haemaphysalis longicornis* as the predominant vector.

SFTSV S segment encodes a nucleocapsid protein (N). Incorrect usages of nucleoprotein (NP) should be corrected through the manuscript.

Reviewer #3:

Remarks to the Author:

The authors use a DNA vaccine approach to evaluate the immunogenicity and protective efficacy of the SFTSV structural and nonstructural proteins. The initial experiments are in mice and involve vaccination with a 5 plasmid combination. Both cellular and humoral responses are measured. Next the authors test the protective efficacy of the vaccine in a ferret SFTS disease model they recently published. The ferret experiments include a 5 plasmid combination vaccine, which protected; a 2 plasmid Gn+Gc envelope protein vaccine, which protected; a 3 plasmid non-envelope protein vaccine, which protected; and a passive transfer (anti-GnGc antibody from DNA vaccinated ferrets, which protected). Overall this paper has several "firsts" and is an important contribution. However there are some issues to address.

Major issues:

- All of the challenges were conducted only two weeks after the last vaccination. This is a very short amount of time and there is a danger that an innate response or nonspecific adaptive response to the vaccine could prevent infection/disease. This would not be as much of a concern if this possibility was controlled for. However, the negative control vaccine (empty vector) does not produce a protein and therefore does not elicit an adaptive immune response. All of the vaccines tested protected. Similarly, passive transfer of immune serum protected; however, nonimmune serum was not injected as a control. It would have been interesting if serum from Ns/NSs/RdRp DNA vaccinated ferrets were tested because if that sera did not protect that would, in effect, serve as a control--- and if it did protect then the possibility that non-neutralizing antibody was playing a role would have been addressed. If there is any way the authors can provide data to rule out the possibility that they are observing innate/non-specific protection, then that should be provided.
- How do we know the Ns, NSs, and RdRp proteins are expressed from the plasmids? There is no mention that the DNA vaccines were tested individually for expression in cell culture or as vaccines in animals. The data indicate that at least one of the DNA vaccines is expressing an antigen, but how do we know they all are? Do the authors have data showing successful expression of those proteins?
- In the abstract one of the conclusions is... "non-envelope-specific T cell responses also can confer protection"... Have non-neutralizing antibodies been ruled out? There was no passive transfer or adaptive transfer from the group vaccinated with non-envelop DNA vaccine plasmids. See line 224 on p. 11. Non-neutralizing antibodies have not been ruled out, although I agree the cellular response is more likely to be the mechanism of protection in the absence of neutralizing antibodies.
- In the conclusion, do the authors consider the 'novel DNA vaccine' the 5 plasmid combination?

Minor issues:

- Second sentence of abstract and the last sentence of the introduction should be reworded.
- The second paragraph says a SFTS model was lacking, but the authors had published the model so that paragraph should not list that as an obstacle (it was an obstacle until they published the model paper).
- Check Order, family, species. Some changes have been made. See Phenuiviridae family.
- Fig. 1e. are the significance *** correct? It appears the difference between control and Vaccine/EP would be more significant. Also, typo in a. reader should be leader.
- The authors switch from intramuscular EP injection in mice to ID EP injection in ferrets. Any reason for this worth commenting on.
- The vaccine constructs are consensus sequences so, technically, all challenges will be "non-homologous." Perhaps using that term is unnecessary.
- P. 9, line177 missing word "with" after "immunized."
- P. 10, line 207. Instead of saying "antibodies are required for protective immunity" it might be more accurate to say "antibodies are sufficient for protective immunity."
- Is the word "middle-sized" needed in line 235, p 11? Or is this the first complete protection in ANY SFTSV challenge using an immunocompetent animal model?
- P. insert volume of serum injected on (p. 14, line 277 method).
- The challenge dose more than 10 million TCID50--- What is the infectious dose (ID50 or ID99) and lethal dose (LD50)? Can that information be provided in the Challenge section of methods?

Comments from the reviewers:

Reviewer #1 (Remarks to the Author):

Kwak et. al demonstrating that using SFTSV-specific DNA vaccine can protect animals which challenged by the SFTSV. Moreover, the authors investigate the immune response activated by this DNA- vaccine. In general, this is well written MS describing the efficacy of DNA vaccine for STFSV. Overall, the approach used in this MS is not novel and the data are not surprising. However, it should be highlighted that data are solid, but the results is of interest for small scientific community (experts in the STFSV and DNA vaccine community) and lack the wider field.

Minor comments:

- **It is interesting to investigate the protection efficacy by just using N and or Ns**

Author response: We appreciate this critical comment. In accordance with the reviewer's suggestion, we investigated the protection efficacy with individual DNA vaccines encoding N or NSs (also RdRp). As shown in revised **Fig. 4** and **Suppl. Fig. 10**, all individual DNA-vaccinated nine ferrets (V39–V41, N-vaccinated ferrets; V42–V44, NSs-vaccinated ferrets; V45–V47, RdRp-vaccinated ferrets) manifested a significantly higher viral load with clinical symptoms (mild thrombocytopenia, high fever, and slight body weight loss) after SFTSV challenge, compared to N/NSs/RdRp-vaccinated ferrets (V35–V38). Notably, one N DNA-vaccinated ferret (V39), one NSs DNA-vaccinated ferret (V42), and one RdRp DNA-vaccinated ferret (V45) survived after lethal SFTSV challenge with complete viral clearance and normal platelet/WBC counts.

Collectively, our results suggest that individual DNA vaccines containing N or NSs (also or RdRp) can contribute to only partial, not complete, protection against SFTSV infection. These new data further reinforce that Gn/Gc would be the most effective antigen for inducing protective immunity. Moreover, vaccines encoding N, NSs, or RdRp may provide an additional protective effect when used in combination with Gn/Gc DNA vaccine as we showed that the inoculation of Gn/Gc DNA vaccines together with non-Env DNA vaccines induced sterilizing immunity in all vaccinated ferrets. We present these data in revised **Fig. 4** and **Suppl. Fig. 10**, describe the results and discuss this important issue in the revised manuscript (page 13 line 279 – page 14 line 292 and page 14 line 306 – page 15 line 312).

- **It would also interesting to see the protection efficacy corresponding to the different amount of Plasmid used (the lowest amount that can give protection).**

Author response: As the reviewer suggested, we performed an additional ferret study to examine the protection efficacy when using vaccines with lower amounts of plasmid DNA. Three ferrets were vaccinated with a total of 200 µg of plasmid DNA (40 µg of each plasmid), and three ferrets were vaccinated with a total of 40 µg of plasmid DNA (8 µg of each plasmid). Strikingly, vaccination with 40 µg of plasmid DNA (V18–V20) induced strong SFTSV-specific T-cell responses, and strong neutralizing activity. These ferrets survived lethal SFTSV challenge without thrombocytopenia, although one of the three ferrets manifested very low viremia at day 2 post-challenge. These results suggest that a vaccination with a small amount of DNA can induce clinically significant SFTSV-specific immune responses that can protect from lethal SFTSV challenge (**Suppl. Fig. 6**). Indeed, ferrets vaccinated with 200 µg of plasmid DNA (V15–V17) were completely protected from lethal SFTSV challenge, with no clinical signs of infection (**Suppl. Fig. 6**). These new data are presented in revised **Suppl. Fig. 6**, and described in the revised manuscript (page 9 line 189 – page 10 line 200).

Thank you again for this helpful comment. These new results are very useful for finding the minimum dose of a DNA vaccine that can induce sufficient immune responses to protect from SFTSV infection in future clinical trial using our DNA vaccine candidates.

- **How many days post challenge he survived animal followed?**

Author response: The survival of challenged ferrets was monitored for 14 days after lethal SFTSV challenge because our recently published study (*Nat Microbiol* 2019; 4(3):438-446) demonstrated that all aged ferrets died within 10 days post-SFTSV challenge. Accordingly, in our current study, all 13 mock-vaccinated control ferrets (C1–C13) died within 10 days post-SFTSV challenge. We recorded clinical signs of infection (viral load, platelet counts, WBC counts, ALT/AST levels, body weight, and body temperature) at 0, 2, 4, 6, and 8 days post-challenge because our previous studies clearly demonstrated that these clinical symptoms were completely resolved in survived vaccinated ferrets within 8 days post-challenge. However, if vaccinated ferrets showed any clinical signs of infection (based on platelet and WBC counts, body temperature, and/or body weight possibly due to partial protection by vaccination), they were monitored for their clinical symptoms for 14 days post-challenge. Even for those ferrets that initially showed minor clinical signs, they ultimately exhibited

no detectible viremia at 14 days post-SFTSV challenge. We have added this information to the revised manuscript (page 21 line 452 – page 22 line 459)

• **How the Viral titers determined in figure2 and 3? Based on PCR or infectious particles?**

Author response: We apologize for the insufficient description. We determined viral titers based on qRT-PCR assay. We have revised a detailed description of the qRT-PCR methods in the “Materials and Methods” section of the revised manuscript (page 22 line 465–473).

Reviewer #2 (Remarks to the Author):

In this manuscript, the authors generate a DNA vaccine against Severe Fever with Thrombocytopenia Syndrome Virus (SFTSV) in a recently-established lethal, aged ferret model which recapitulates the clinical manifestations seen in SFTS patients. Following three vaccinations with a mixture of five DNA vaccines (against Gn, Gc, NP, RdRp, and NSs) at 2-week intervals, ferrets were lethally challenged with SFTSV. DNA vaccination conferred complete protection in the lethal, aged ferret model, while control ferrets succumbed to SFTSV infection. The authors further demonstrate the Gn/Gc are important for neutralizing antibody response in the ferret model, while the NSs/NP/RdRp elicit a T-cell specific response. While these studies show protection from lethal SFTSV infection in a novel animal model, there are major concerns that should be considered for this manuscript.

Major Concerns

While the authors present a platform for an SFTSV vaccine, a study was recently published using a single-dose of an rVSV-SFTSV/AH12-GP that protects IFNAR^{-/-} mice from lethal SFTSV infection (Dong F, et al. *npj Vaccines*. 2019. doi: 10.1038/s41541-018-0096-y). The authors of the present study state that their “vaccine is the first to demonstrate complete protection against lethal SFTSV challenge using an immunocompetent, middle-size animal model with clinical manifestations of SFTSV infection”, which may be misleading to the reader. The authors should refer to the previous related study and tone down the sentence above.

Author response: We appreciate this comment. We have added the reference and toned down the sentence in the revised manuscript (page 14 line 300–306)

“Although it was recently reported that a live attenuated recombinant vesicular stomatitis virus (rVSV)-based vaccine expressing the SFTSV Gn/Gc glycoproteins can elicit protection against SFTSV in immunocompromised IFNAR^{-/-} mice (NPJ Vaccines 2019; 4:5), our current study represent a substantial contribution towards the development of an effective preventive vaccine for SFTS, considering that our SFTSV DNA vaccines induced complete protection against lethal SFTSV challenge in an immunocompetent middle-sized animal model that exhibits clinical manifestations seen in SFTS patients (Nat Microbiol 2019; 4(3):438-446).”

Furthermore, while the DNA vaccine does elicit protective immunity in the animal model presented, this is only following three vaccinations, which may not be realistic in a clinical setting. Have the authors observed protection or partial protection in ferrets vaccinated

once?

Author response: We appreciate this helpful comment. As the reviewer suggested, we performed a new ferret study to examine the protection efficacy after vaccination once or twice. All ferrets vaccinated twice (V9–V11) survived without developing any clinical signs of infection after lethal SFTSV challenge (**Suppl. Fig. 5**), indicating that two vaccinations were sufficient for the induction of protective immunity against SFTSV infection. However, ferrets vaccinated once (V12–V14) manifested significant viremia and high fever after SFTSV challenge and two out of three ferrets died after lethal challenge. Notably, one ferret (V13) showed that a single vaccination still induced strong SFTSV-specific T-cell responses. This ferret survived from lethal SFTSV challenge and was completely cleared of virus at 14 days post-challenge (**Suppl. Fig. 5**).

Collectively, these results suggest that twice DNA vaccination appears to be sufficient for complete protection against lethal SFTSV infection, and that single DNA vaccination achieves only partial protection. We present these data in revised **Suppl. Fig. 5**, and describe the results in the revised manuscript (page 9 line 174–188).

The authors previously established a lethal immunocompetent animal model for the study of SFTSV using outbred, aged ferrets, in which aged ferrets lethally infected with SFTSV showed clinical signs of SFTS infection: thrombocytopenia, leukopenia, fever, and elevated AST/ALT. The current manuscript does not show these additional clinical signs in their vaccinated animals or control animals. Can the authors provide rationale for leaving out these experiments in the present manuscript, which have been previously performed in the same animal model? This would strengthen the efficacy of their vaccine platform.

Author response: We appreciate this helpful comment. In the original version of the manuscript, we mainly presented % survival rates, viral titers, and platelet counts because they seemed to be the most reliable representative parameters for evaluating clinical signs of infection. Other clinical signs of infection—including WBC counts, fever, body-weight changes, and AST/ALT levels—were also evaluated throughout the study. In accordance with the reviewer’s suggestion, we have included these data (**Suppl. Fig. 4–7 and 9–10**) and described these results in the revised manuscript (page 9 line 167, page 9 line 172–173, page 11 line 227–229, page 12 line 244–245, page 13 line 275–276, and page 14 line 289–290).

Immunization with the mixture of five different DNA vaccine plasmids is unusual and not explained in the text. The rationale of this vaccination strategy should be provided in the

manuscript. This vaccination scheme does not help to define the most effective antigen to induce protective immunity in vaccinated animals. Moreover, the authors should perform the protection studies with individual DNA vaccine which encode the NSs, N, or RdRp. In general, immunization with nucleocapsid protein as an antigen elicits T cell-mediated protective immunity. Interestingly, in Figure 1d, vaccination potentially elicit T cell-mediated immune response against NSs. Therefore, to determine which antigen (N or NSs) is effective antigen to induce protective cell-mediated immunity, the authors should perform protection studies in the ferrets vaccinated with N or NSs DNA vaccine.

Author response: We thank reviewer #2 for this insightful comment. Prior to our current study, no study has described the most effective antigen to induce protective immunity in vaccinated animals. For this reason, we initially vaccinated ferrets with a mixture of five different DNA vaccine plasmids expressing all SFTSV proteins to investigate whether our DNA vaccine platform would be an effective means to induce protective immunity against lethal SFTSV infection,. In this revision, in accordance with the reviewer's suggestion, we performed protection studies using individual DNA vaccine encoding N, NSs, or RdRp to define the most effective protective antigen.

As shown in revised **Fig. 4** and **Suppl. Fig. 10**, all individual DNA-vaccinated nine ferrets (V39–V41, N-vaccinated ferrets; V42–V44, NSs-vaccinated ferrets; V45–V47, RdRp-vaccinated ferrets) manifested a significantly higher viral load with clinical symptoms (mild thrombocytopenia, high fever, and slight body weight loss) after SFTSV challenge, compared to N/NSs/RdRp-vaccinated ferrets (V35–V38). Notably, one N DNA-vaccinated ferret (V39), one NSs DNA-vaccinated ferret (V42), and one RdRp DNA-vaccinated ferret (V45) survived after lethal SFTSV challenge with complete viral clearance and normal platelet/WBC counts.

Collectively, our results suggest that individual DNA vaccines containing N, NSs, or RdRp can contribute to only partial, not complete, protection against SFTSV infection. These new data further reinforce that Gn/Gc would be the most effective antigen for inducing protective immunity. Moreover, vaccines encoding N, NSs, or RdRp may provide an additional protective effect when used in combination with Gn/Gc DNA vaccines as we showed that the inoculation of Gn/Gc DNA vaccines together with non-Env DNA vaccines induced sterilizing immunity in all vaccinated ferrets. We present these data in revised **Fig. 4** and **Suppl. Fig. 10**, describe the results and discuss this important issue in the revised manuscript (page 13 line 279 – page 14 line 292 and page 14 line 306 – page 15 line 312).

The authors investigated “the role of the anti-envelope antibody response as a protective immunity against SFTSV infection” by transferring sera from Gn/Gc vaccinated ferrets to

naive ferrets. While this experimental outline and rationale are clear, the authors lack an experimental control for these studies; for the results presented, serum-transferred ferret survival rates, viral loads, and platelet counts are directly compared to control-vaccinated ferrets. Control vaccinated ferret serum transfer to naïve ferrets should be done in parallel with Gn/Gc vaccinated animal serum transfer to naïve animals. Furthermore, the authors did not comment on any other clinical disease signs in vaccinated or serum transfer animals (fever, leukopenia, AST/ALT) even though they had low viral titers.

Author response: We appreciate this critical comment. In accordance with the reviewer's suggestion, we investigated the protective efficacy of treatment with serum from pVax1(empty vector)-vaccinated ferrets against lethal SFTSV challenge. As shown in revised **Fig. 3e and Suppl. Fig 8–9**, all three ferrets that received pVax1-vaccinated ferrets (V25–V27) died within 10 days post-lethal SFTSV challenge with typical clinical signs of SFTS. We describe the results in the revised manuscript (page 12, line 246–254).

Additionally, we found that all three ferrets that received serum from N/NSs/RdRp-vaccinated ferrets (V32–V34) exhibited severe clinical signs of SFTS and died after lethal SFTSV challenge. This suggests that the anti-envelope antibody response plays a critical role in the protective immunity against SFTSV infection. We present these data in revised **Fig. 3e and Suppl. Fig 8–9**, and describe the results in the revised manuscript (page 12, line 246–257).

In accordance with the reviewer's comment, we have also added data regarding other clinical disease signs in Gn/Gc-vaccinated and serum-transferred ferrets—including fever, thrombocytopenia, WBC counts, and ALT/AST levels—in the revised manuscript (**Suppl. Fig. 7 and 9** and page 11 line 227–229, page 12 line 244–245, and page 12 line 252–254).

Minor Concerns:

Line 59-61: According to current ICTV classification, SFTSV is a member of the genus Bangyangvirus in the family Phenuiviridae of the order Bunyvirales. The sentence should be corrected with correct taxonomical classification.

Author response: In accordance with the reviewer's comment, we have corrected this sentence (page 3 line 40–41, and page 4 line 60–63).

Line 61-62: The authors write that the “the L segment encodes the viral RNA polymerase”. This should be changed to the “viral RNA-dependent RNA polymerase”.

Author response: In accordance with the reviewer’s comment, we have corrected this sentence (page 4, line 63–64).

“the L segment encodes the viral RNA-dependent RNA polymerase”

Line 63-64: The authors write “SFTSV is an arbovirus transmitted by the tick Haemaphysalis longicornis”, however, Rhipicephalus microplus, and others are vectors of SFTSV (Yun, SM. Emerg Infect Dis 2014. doi: 10.3201/eid2008.131857). The authors should consider re-writing this sentence to include other vectors, or may want to indicate Haemaphysalis longicornis as the predominant vector.

Author response: In accordance with the reviewer’s comment, this sentence has been revised to “SFTSV is an arbovirus transmitted by the *Haemaphysalis longicornis* tick as the predominant vector, as well as by the *Rhipicephalus microplus* tick and others” (page 4, line 66–67).

SFTSV S segment encodes a nucleocapsid protein (N). Incorrect usages of nucleoprotein (NP) should be corrected through the manuscript.

Author response: In accordance with the reviewer’s comment, we have changed “NP” to “N” throughout the entire revised manuscript.

Reviewer #3 (Remarks to the Author):

The authors use a DNA vaccine approach to evaluate the immunogenicity and protective efficacy of the SFTSV structural and nonstructural proteins. The initial experiments are in mice and involve vaccination with a 5 plasmid combination. Both cellular and humoral responses are measured. Next the authors test the protective efficacy of the vaccine in a ferret SFTS disease model they recently published. The ferret experiments include a 5 plasmid combination vaccine, which protected; a 2 plasmid Gn+Gc envelope protein vaccine, which protected; a 3 plasmid non-envelope protein vaccine, which protected; and a passive transfer (anti-GnGc antibody from DNA vaccinated ferrets, which protected. Overall this paper has several “firsts” and is an important contribution. However there are some issues to address.

Major issues:

- All of the challenges were conducted only two weeks after the last vaccination. This is a very short amount of time and there is a danger that an innate response or nonspecific adaptive response to the vaccine could prevent infection/disease. This would not be as much of a concern if this possibility was controlled for. However, the negative control vaccine (empty vector) does not produce a protein and therefore does not elicit an adaptive immune response. All of the vaccines tested protected.

Author response: We appreciate this critical comment. As suggested, we performed a new ferret study to investigate the protection efficacy when vaccinated ferrets were challenged at ‘4 weeks’ after their last vaccination. Similar to in the previous ferret challenge studies, all vaccinated ferrets (V5–V8) that were challenged with lethal SFTSV at 4 weeks after their last vaccination survived without exhibiting any clinical signs of SFTS (revised **Fig. 2** and **Suppl. Fig. 4**). Moreover, the ferrets challenged either at 2 weeks or at 4 weeks after the last vaccination showed similar % survival rate, serum viral load, platelet counts, WBC counts, temperature changes, and % relative body weight (**Suppl. Fig. 4**). These results suggest that an innate response or nonspecific adaptive response may not play a critical role in immune protection.

We present these data in the revised manuscript (revised **Fig. 2** and **Suppl. Fig. 4**) and describe the results in the revised manuscript (page 8 page 156–157, and page 9 line 170–173).

Similarly, passive transfer of immune serum protected; however, nonimmune serum was not injected as a control. It would have been interesting if serum from Ns/NSs/RdRp DNA vaccinated ferrets were tested because if that sera did not protect that would, in effect,

serve as a control--- and if it did protect then the possibility that non-neutralizing antibody was playing a role would have been addressed. If there is any way the authors can provide data to rule out the possibility that they are observing innate/non-specific protection, then that should be provided.

Author response: We appreciate this critical comment. In accordance with the reviewer's suggestion, we performed two new ferret studies to investigate transfers of nonimmune serum or serum from N/NSs/RdRp DNA-vaccinated ferrets. As shown in revised **Fig. 3e and Suppl. Fig 8–9**, all three ferrets that received nonimmune serum from pVax1(empty vector)-vaccinated ferrets (V25–V27) exhibited severe clinical signs of SFTS and died after lethal SFTSV challenge. Moreover, all three ferrets that received serum from N/NSs/RdRp-vaccinated ferrets (V32–V34) also died within 11 days post-lethal SFTSV challenge with typical clinical signs of severe SFTS.

These results indicate that non-neutralizing antibody does not induce protective immunity. In the revised manuscript, we present these data (**Fig. 3e and Suppl. Fig 8– 9**) and describe the results (page 12, line 246–254).

- How do we know the Ns, NSs, and RdRp proteins are expressed from the plasmids? There is no mention that the DNA vaccines were tested individually for expression in cell culture or as vaccines in animals. The data indicate that at least one of the DNA vaccines is expressing an antigen, but how do we know they all are? Do the authors have data showing successful expression of those proteins?

Author response: In accordance with the reviewer's critique, we performed western blot assays to examine the expressions of Gn, Gc, N, NSs, and RdRp. In the revised manuscript, we present these data (revised **Fig. 1b**), and describe the methods and the results (page 6, line 100–101, and page 16 line 329–338).

To further support this, we immunized BALB/c mice with individual DNA vaccine encoding Gn, Gc, N, NSs, or RdRp, and performed IFN- γ ELISPOT assays. Indeed, We observed that all five individual DNA vaccine elicited strong T-cell immunity specific to corresponding antigen in mice. We present these data (**Suppl. Fig. 2a**), and describe the results in the revised manuscript (page 6, line 111–114).

- In the abstract one of the conclusions is... “non-envelope-specific T cell responses also can confer protection”... Have non-neutralizing antibodies been ruled out? There was no passive transfer or adaptive transfer from the group vaccinated with non-envelop DNA

vaccine plasmids. See line 224 on p. 11. Non-neutralizing antibodies have not been ruled out, although I agree the cellular response is more likely to be the mechanism of protection in the absence of neutralizing antibodies.

Author response: As described above, we investigated the protection efficacy of serum from non-envelope (N/NSs/RdRp) DNA-vaccinated ferrets against lethal challenge (**Fig. 3e** and **Suppl. Fig 8–9**). All ferrets that received serum from N/NSs/RdRp DNA-vaccinated ferrets died within 11 days after SFTSV challenge and exhibited typical clinical signs of SFTS prior to death. This suggests that non-neutralizing antibody specific to non-envelope antigens does not induce protection against SFTSV challenge. In the revised manuscript, we present these data (**Fig. 3e** and **Suppl. Fig 8–9**) and discuss this issue (page 12, line 246–254).

- In the conclusion, do the authors consider the ‘novel DNA vaccine’ the 5 plasmid combination?

Author response: As requested by reviewer #1 and #2, for the revised manuscript, we performed new ferret studies to investigate the protection efficacy of individual DNA vaccines encoding N, NSs, or RdRp (the protection efficacy of the Gn/Gc DNA vaccine was investigated in the original study). As shown in revised **Fig. 4** and **Suppl. Fig. 9**, our new data suggest that the N, NSs, or RdRp DNA vaccine alone may not be sufficient for protection against SFTSV infection. In contrast, ferrets vaccinated with Gn/Gc DNA vaccines were completely protected from lethal non-homologous SFTSV challenge, and developed no clinical signs.

These findings strongly indicate that the DNA vaccine encoding Gn/Gc is the most important component of the novel DNA vaccine. Notably, DNA vaccines encoding N, NSs, or RdRp may provide additional protection effects when combined with Gn/Gc DNA vaccine. The revised manuscript includes discussion of this issue (page 14 line 306 – page 15 line 315).

Minor issues:

- Second sentence of abstract and the last sentence of the introduction should be reworded.

Author response: In accordance with the reviewer’s comment, these sentences have been revised (page 3, line 41–42, and page 5 line 90–93).

- The second paragraph says a SFTS model was lacking, but the authors had published the

model so that paragraph should not list that as an obstacle (it was an obstacle until they published the model paper).

Author response: In accordance with the reviewer's comment, we have revised these sentences (page 4, line 77 – page 5 line 86).

- Check Order, family, species. Some changes have been made. See Phenuiviridae family.

Author response: In accordance with the reviewer's comment, sentences has been revised to “*SFTS virus (SFTSV) belonging to the genus Bangyangvirus in the family Phenuiviridae of the order Bunyavirales*” (page 3 line 40–41, and page 4 line 60–63).

- Fig. 1e. are the significance * correct? It appears the difference between control and Vaccine/EP would be more significant. Also, typo in a. reader should be leader.**

Author response: We apologize for the mistake. Significance between ‘Vaccine’ and ‘Vaccine/EP’ should be ‘***’, not ‘*****’. We have corrected this error in the revised **Fig. 1e**.

- The authors switch from intramuscular EP injection in mice to ID EP injection in ferrets. Any reason for this worth commenting on.

Author response: Pre-clinical evaluation of DNA plasmids can utilize either IM or ID delivery, depending on the species. In contrast, in humans, ID delivery is preferred due to the reduced pain and greater ease of delivery. More importantly, studies in NHPs, rabbits, and humans have recently shown that ID delivery of DNA vaccines yields equal or greater immunogenicity and protection (*Vaccine* 2016; 34(31):3607-12, *Hum Gene Ther Methods* 2015; 26(4):134-46, *Vaccines* 2013; 1(3):262-77, and *J Infect Dis* 2019; doi: 10.1093/infdis/jiz132).

In the current study, we used ID/EP injection because ferrets have thicker epidermal and dermal layers that enable ID delivery, while ID delivery is not possible in mice. We are considering ID/EP injection for a future clinical trial. We discuss this issue in the revised manuscript (page 17, line 359–363).

- The vaccine constructs are consensus sequences so, technically, all challenges will be “non-homologous.” Perhaps using that term is unnecessary.

Author response: We agree with the reviewer’s suggestion, and have deleted the word “non-homologous” throughout the revised manuscript.

- P. 9, line177 missing word “with” after “immunized.”

Author response: In accordance with the reviewer’s comment, we have revised this mistake (page 10, line 212–213).

- P. 10, line 207. Instead of saying “antibodies are required for protective immunity” it might be more accurate to say “antibodies are sufficient for protective immunity.”

Author response: In accordance with the reviewer’s comment, this sentence has been changed to “... *antibodies are sufficient for protective immunity ...*” (page 12, line 255–256).

- Is the word “middle-sized” needed in line 235, p 11? Or is this the first complete protection in ANY SFTSV challenge using an immunocompetent animal model?

Author response: To our knowledge, our study is the first study to demonstrate complete protection by vaccination against a lethal SFTSV challenge using an ‘immunocompetent’ animal model. We believe that the evaluation of vaccine protection efficacy in a middle-sized animal (not a mouse model) is one of the novelty of the current study. Therefore, we would like to keep these words in the manuscript. However, in accordance with the reviewer’s comment, we have revised the manuscript (page 14, line 302–306).

“our SFTSV DNA vaccines induced complete protection against lethal SFTSV challenge in an immunocompetent middle-sized animal model that exhibits clinical manifestations seen in SFTS patients.”

- P. insert volume of serum injected on (p. 14, line 277 method).

Author response: In accordance with the reviewer’s comment, we now describe the volume of injected serum in the revised manuscript (page 11, line 233–234, and page 12, line 247–248).

- The challenge dose more than 10 million TCID50--- What is the infectious dose (ID50 or ID99) and lethal dose (LD50)? Can that information be provided in the Challenge section of methods?

Author response: We thank the reviewer for this comment. The infectious dose (ID₅₀ or ID₉₉) and lethal dose (LD₅₀) have not yet been fully determined since our ferret model of lethal SFTSV infection has been recently established. We would like to stress that unlike laboratory mice, ferrets are outbred middle-sized animal with different genetic backgrounds. Particularly, aged (>4 years old) ferrets are not easily found in large number and consequently of high value. Determination of the ID₅₀ and LD₅₀ values requires many naïve young and aged ferrets, which may be practically and ethically difficult. In fact, our Institutional Animal Care and Use Committees (IACUC) has not approved these. Thus, we plan to continuously accumulate and subsequently extrapolate our experimental data to determine the ID₅₀ and LD₅₀ values rather than scarify a hundred of naïve young and aged ferrets to solely the ID₅₀ and LD₅₀ values.

Nevertheless, our main goal in the current study is to demonstrate the DNA vaccines' protective efficacy against lethal challenge of SFTSV infection in vaccinated ferrets. Therefore, we used ~10^{7.6} TCID₅₀ dose of SFTSV throughout the ferret challenge studies, which caused fatal infection in 100% of ferrets in our previous recent study (*Nat Microbiol* 2019; 4(3):438-446). Our current study also showed that 100% of mock-vaccinated ferrets (13/13) (control group) died within 10 days post-challenge when infected with 10^{7.6} TCID₅₀ of SFTSV. We have discussed this issue in the revised manuscript (page 21 line 450–452).

Reviewers' Comments:

Reviewer #1:

None

Reviewer #2:

Remarks to the Author:

The authors have done a thorough job of revising the present manuscript and their replies to the reviewer's comments are reasonable.

Reviewer #3:

Remarks to the Author:

The authors performed several experiments in response to my comments. I'm satisfied with this response.